# Coronary “Microvascular Dysfunction”: Evolving Understanding of Pathophysiology, Clinical Implications, and Potential Therapeutics

**DOI:** 10.3390/ijms241411287

**Published:** 2023-07-10

**Authors:** Chun Yeung Kei, Kuljit Singh, Rustem F. Dautov, Thanh H. Nguyen, Yuliy Y. Chirkov, John D. Horowitz

**Affiliations:** 1Department of Medicine, University of Adelaide, Adelaide 5371, Australia; kei.jonathan@yahoo.com.hk (C.Y.K.); thanh.h.nguyen@adelaide.edu.au (T.H.N.); yuliy.chirkov@adelaide.edu.au (Y.Y.C.); 2Department of Medicine, Griffith University, Southport 4111, Australia; kuljit.singh@health.qld.gov.au; 3Gold Coast University Hospital, Gold Coast 4215, Australia; 4Department of Medicine, University of Queensland, Woolloongabba 4102, Australia; rustem.dautov@health.qld.gov.au; 5Prince Charles Hospital, Brisbane 4032, Australia; 6Northern Adelaide Local Health Network, Adelaide 5000, Australia; 7Basil Hetzel Institute for Translational Research, Adelaide 5011, Australia

**Keywords:** coronary microvessels, stable angina pectoris, glycocalyx, platelet aggregation, mast cells, vasodilator autacoids, vascular ageing

## Abstract

Until recently, it has been generally held that stable angina pectoris (SAP) primarily reflects the presence of epicardial coronary artery stenoses due to atheromatous plaque(s), while acute myocardial infarction (AMI) results from thrombus formation on ruptured plaques. This concept is now challenged, especially by results of the ORBITA and ISCHEMIA trials, which showed that angioplasty/stenting does not substantially relieve SAP symptoms or prevent AMI or death in such patients. These disappointing outcomes serve to redirect attention towards anomalies of small coronary physiology. Recent studies suggest that coronary microvasculature is often both structurally and physiologically abnormal irrespective of the presence or absence of large coronary artery stenoses. Structural remodelling of the coronary microvasculature appears to be induced primarily by inflammation initiated by mast cell, platelet, and neutrophil activation, leading to erosion of the endothelial glycocalyx. This leads to the disruption of laminar flow and the facilitation of endothelial platelet interaction. Glycocalyx shedding has been implicated in the pathophysiology of coronary artery spasm, cardiovascular ageing, AMI, and viral vasculitis. Physiological dysfunction is closely linked to structural remodelling and occurs in most patients with myocardial ischemia, irrespective of the presence or absence of large-vessel stenoses. Dysfunction includes the impairment of platelet and vascular responsiveness to autocidal coronary vasodilators, such as nitric oxide, prostacyclin, and hydrogen sulphide, and predisposes both to coronary vasoconstriction and to a propensity for microthrombus formation. These findings emphasise the need for new directions in medical therapeutics for patients with SAP, as well as a wide range of other cardiovascular disorders.

## 1. Introduction

Myocardial ischaemic pain, whether precipitated by exertion (stable angina pectoris; SAP) or occurring at rest (for example, associated with acute myocardial infarction (AMI)) reflects imbalanced or impaired myocardial energetics due to an impairment of the coronary flow relative to myocardial oxygen demand. SAP, distinct from AMI (which is, by definition, associated with the induction of myocardial necrosis), usually carries a relatively benign prognosis regarding patient survival but is often a debilitating condition, which imposes a markedly negative impact on the quality of life.

The common clinical view of the pathogenesis of SAP over at least the past 50 years has been that it results from the presence of flow-limiting large-vessel or epicardial coronary stenoses, with the resting coronary flow remaining unchanged by an autoregulation process of the “compensatory” dilatation of the coronary microcirculation but at the expense of the progressive depletion of the coronary vasodilator reserve until eventually angina may occur on minimal exertion or at rest (Class IV angina). This concept of the depletion of the coronary vasodilator reserve has been strongly influenced by the pioneering work of Dr Lance Gould and colleagues, beginning nearly 50 years ago [1]. Gould demonstrated, mainly in animal models of coronary artery stenosis, that the coronary flow reserve, defined as the ratio of maximal to resting flow, progressively diminishes as the degree of large coronary stenosis increases. Essentially, according to this original model, the resultant limitation of the vasodilator reserve due to large-vessel stenosis is the principal basis for angina during exercise, as shown schematically in Figure 1.

In this regard, it is important to understand that coronary circulation can be regarded as a two-component model consisting of epicardial coronary arteries and the coronary microcirculation. The coronary microcirculation is the major site of regulation of flow resistance and, thus, coronary perfusion, while the epicardial coronaries behave more like conduits for blood flow [3].

## 2. Insights from Periprocedural Physiological Studies

It must also be noted that the bulk of Gould’s data is based on positron emission tomography (PET)-derived total coronary flow reserve measurement, which precludes a separate assessment of changes in coronary microcirculation versus large epicardial coronary vessel resistance [1]. On the basis of the theory that ischaemic symptoms are predicated on the severity of large coronary artery stenoses, invasive coronary revascularisation decisions, whether for percutaneous coronary stenting or coronary bypass surgery, have primarily reflected the management of “flow-limiting” epicardial coronary stenoses. However, over the past 30 years, essentially to ensure that percutaneous coronary interventions (PCIs) were performed only on coronary lesions of physiological significance, a number of means of assessment of their imposition on coronary flow were developed to be used as adjuncts to decision making [4]. 

Of these various physiological investigations, “Fractional Flow Reserve” (FFR), a form of evaluation of the impact of transient hyperaemia, has become the most utilised investigation. Essentially, FFR is largely a misnomer, as strictly speaking only the trans-stenotic pressure gradient is measured, usually only prior to a potential PCI. Theoretically, FFR measurement is an assessment of the contribution of a particular coronary stenosis to the impairment of the local coronary haemodynamic reserve, as measured at rest and during pharmacologically induced hyperaemia (either with intravenous infusion or a bolus of intracoronary injection of adenosine) [5,6]. The overall idea behind this is that the demonstration of a substantial pressure gradient across a particular large-vessel stenosis implies that the presence of the stenosis compromises adenosine-induced coronary vasodilatation and, therefore, is of physiological and clinical significance. Of course, adenosine is only one of a large number of independent or synergistically acting endogenous coronary vasodilators.

In theory, given that the pressure gradient measurement for FFR calculation is purely trans-stenotic, the results of FFR determination should be independent of putative small coronary dysfunction. Similarly, the failure of FFR to normalise post-PCI should theoretically be an indication of an inadequate PCI or the diffuse nature of epicardial coronary disease rather than of the presence of microvascular disease and, indeed, that seems to be largely the case [7]. In theory, it would be preferable for the actual flow measurement to occur, and the relative arguments for this, versus the surrogate of the trans-stenotic pressure gradient, have recently been reviewed by Howard and Murthy [8]. Indeed, it turns out that the correlation between FFR and actual coronary flow reserve is not good in the clinical setting [9] and even that FFR may be deceptively increased in the presence of elevated microvascular resistance [9]. Overall, therefore, FFR is a flawed index of large coronary lesion severity: its greatest conceptual flaw is the assumption that the vasodilator response to adenosine within the small coronary vasculature remains constant.

Despite these substantial theoretical limitations, a number of investigations have been performed to test the clinical utility of the measurement of FFR as a decision-making tool to refine the precision and safety of PCIs. The most important of these are the FAME studies, and it is appropriate to review their design and results.

The FAME 1 study [6] was an investigation of the utility of FFR measurement in patients with SAP who, at diagnostic angiography, were found to have multivessel coronary artery stenoses. Patients were randomised to either (visual only) angiography-guided or to FFR-guided PCIs, with FFR < 0.80 representing the criterion for “significant” stenosis and, therefore, for the performance of PCIs within that artery. The FFR measurement resulted in a substantial reduction in the mean number of stents inserted per patient. The primary outcome for FAME-1 was the composite of death, acute myocardial infarction, and repeat revascularisation. One-year post-PCI, there was a significantly lower rate of this endpoint in patients who had been randomised to the FFR-guided PCI, driven mainly by the rates of myocardial infarction and of repeat revascularisation. It was suggested that the benefit of FFR guidance might have been mediated by the elimination of unnecessary PCI procedures, but, on the other hand, the period of follow-up was short, limiting the assessment of the impact of lesion progression. The FAME 1 study results complemented those of the earlier DEFER study, where the deferral of the PCI of a functionally nonsignificant stenosis, defined by FFR >0.75, was associated with a favourable very long-term follow-up at 15 years without signs of the late “catch-up” phenomenon, and the outcomes did not change poststenting [10,11].

The next major report in this series was the FAME-2 study [5], which compared FFR-guided PCI results with those of long-term medical therapy alone. A total of 888 patients were randomised within this trial design before recruitment was stopped prematurely due to differences between groups as regards the primary endpoint of the study: a composite death, AMI, and *urgent* revascularisation. It is important to recognise that these results were driven entirely by a 7-fold difference in rates of urgent revascularisation [5]. There were very few deaths or myocardial infarctions within the initial 12-month follow-up period. At the time of a review after a 2-year follow-up of this patient cohort [12], the composite endpoint results remained different, again entirely driven by differences in the rates of urgent revascularisation. However, there was also a 5-fold higher rate of nonurgent revascularisation among medically treated patients. Finally, after the 5-year follow-up [13], there was still a halving of the composite endpoint risk, with a borderline-significant reduction in rates of AMI.

The results of the FAME studies, especially those of FAME-2, have been mired in controversy, and the uptake of FFR-guided therapy outside the United States has been low. Much of the criticism has been engendered by the combination of the nonblinded nature of the trials, combined with the impact of potentially biased decision-making as to which patients merited “urgent revascularisation” [14]. However, other points of criticism included the very small number of “hard” endpoints in these trials, the lack of assessment of actual coronary blood flow, the low-risk status of the overall population studied, and the exclusion of patients with periprocedural AMI from the overall data analysis, despite the fact that some of these AMI cases may well have been of prognostic importance [15].

## 3. ORBITA and ISCHEMIA Studies: Efficacy of PCI 

In a sense, a counterpoint to the thrust of the FAME-1 and FAME-2 studies was reached with the publication of the ORBITA [16] and ISCHEMIA [17] studies in 2018 and 2020, respectively. These studies focused on the actual benefits of myocardial revascularisation (mainly via PCIs) on the symptomatic status [16] and risk of AMI/death [17], respectively. The results of these studies have led to substantial changes in clinical decision-making for patients with SAP.

The ISCHEMIA trial [17] evaluated more than 5000 patients with inducible myocardial ischemia (on stress testing plus physiological imaging in most cases), plus epicardial coronary stenoses. Despite these inclusion criteria, most patients had only mild SAP. The result was startling. Over an approximately 2.5-year, then 3.2-year [18] follow-up, the revascularisation group (74% PCI, 26% bypass surgery) had similar rates of the primary composite endpoint (death, myocardial infarction, hospitalisation for unstable angina, congestive heart failure, and resuscitated cardiac arrest) as those for patients treated with ongoing medical measures alone (see Table 1). Notably, the risk of the primary endpoint was slightly higher in the first six months in the invasive group but lower than in the conservative-treated group at the end of the average 2.5-year follow-up period, raising the possibility of long-term net benefits essentially overshadowing periprocedural harm. A subsequent analysis went further: it was shown that the long-term significance of what was designed as a “procedural infarct” was less than that of a spontaneous infarct [19], suggesting that these two categories of AMI should be treated somewhat differently as regards the significance of postprocedural outcomes. This analysis supports findings of the ACUITY trial that the long-term impact of periprocedural MI is not similar to spontaneous MI [20].

The primary endpoint consisted of any of the following: cardiovascular death, AMI, unstable angina, heart failure and resuscitated cardiac arrest;Heart failure and unstable angina were only considered if they led to hospitalisation.

In terms of symptomatic relief in SAP patients following PCI, ORBITA [19] represents a landmark single-blinded study, including 200 patients. Patients, generally with SAP of moderate severity, who had coronary lesions suitable for a PCI and FFR <0.8, were randomly assigned to receive a real PCI or pseudo-PCI. Exercise tolerance was marginally (and nonsignificantly) higher postrecovery in the actual PCI group. Therefore, the ORBITA results showed that for patients with SAP of a moderate severity, a PCI offers a minimal symptomatic benefit over that of conventional medical therapy. As regards the interaction between the individual FFR data and symptomatic changes post-PCI, the FFR data did not predict changes in exercise time post-PCI [21].

In both the ORBITA and ISCHEMIA trials, patients with severe angina tended to be excluded. Thus, the results of both ORBITA and ISCHEMIA relate primarily to the (poor) results of invasive interventions relative to medical treatment in SAP patients, who are mildly symptomatic. Nevertheless, it is counterintuitive that an intervention which reduces large coronary artery resistance should not improve symptomatic status. 

In summary, the results from the above studies indicate that the current invasive treatments for large coronary vessel stenoses, especially PCIs, are, at best, minimally effective in improving SAP symptoms and have no medium-term prognostic benefits. It must be noted that the data regarding revascularisation by coronary bypass surgery within the ISCEMIA trial are limited, so no definitive conclusions regarding the extent of the benefits can be reached.

The results of ORBITA and ISCHEMIA may also indicate that the previous Gould- inspired concept of ischaemic physiology based entirely on the impact of the severity of large-vessel stenosis is oversimplified. It had been assumed that the impact of microvascular components in coronary autoregulation is relatively constant and, thus, that the coronary flow reserve is compromised entirely by the degree of the stenosis in the corresponding large coronary vessel. Many recent studies [22,23] have suggested that the lack of the predictive accuracy of FFR measurement and the relatively minor benefits of PCIs both reflect the presence and variable impact of “small vessel dysfunction” as an important modulator of coronary flow reserve and proposed that current revascularisation (and indeed medical therapy) fails to address a major component of the pathogenesis of SAP. However, details of the precise nature of such “small vessel dysfunction” remain sparse. 

## 4. Assessment of Coronary Microcirculation

Unlike epicardial coronary vessels, the small dimension of the coronary microvasculature prevents its direct visualisation. Therefore, in the last 30 years, attempts have been made to assess coronary microcirculatory function instead. 

Cardiac PET and cardiac magnetic resonance imaging (CMR) have been the major noninvasive assessment tools used to measure CFR. The assessment of CFR using PET and CMR provides independent prognostic information on patients’ coronary disease and SAP [23,24,25]. However, it is difficult to assess the two components of coronary circulation separately with noninvasive tools.

Invasively, both coronary angiography and pressure wire measurements have been used to assess microcirculatory function. Figure 2 details invasive tests of large and small coronary vascular function, which have been utilised in clinical investigations. A major positive of the invasive measurement of coronary physiology is the ability to assess the tw components of the model of coronary circulation separately. The index of coronary microvascular resistance (IMR) is likely to be the most precise, reproducible, and least variable invasive measure of the coronary microcirculatory function. IMR is calculated with thermodilution as the product of the distal coronary pressures (Pd) and the mean transit time during maximal hyperaemia (if the venous pressure is close to zero) [22]. IMR values were tested in patients presenting for an angiography for noncoronary disease indications. The normal range of IMR is <25 based on three studies reporting the IMR value in healthy populations [26,27,28]. A value of ≥25 suggests high microvascular resistance and, hence, coronary microcirculatory dysfunction [29]. Subsequently, to simplify the measurement of IMR and to remove the restrictions related to wiring the coronary artery, an angiography-guided IMR-Angio was developed and validated against invasive IMR and CMR [30]. Impaired IMR-Angio as a measure of microcirculatory dysfunction when assessed post-PCI both in SAP and AMI cohorts has been found to be a marker of worse outcomes. Dai et al. demonstrated that during a median of a 28-month follow-up after an index PCI procedure, patients with IMR of ≥25.1 demonstrated a significantly higher incidence of cardiac death or readmission due to heart failure than those with IMR <25.1 [31]. Similarly, in an AMI cohort, post-PCI higher values of IMR have been shown to be associated with a significantly higher risk for cardiac death or readmission for heart failure than those with preserved IMR values [32]. 

The corrected TIMI frame count (CTFC) is an objective and quantitative angiographic method, where the number of cine frames needed for the contrast dye to run off from the coronary artery can be used to indirectly assess microcirculatory dysfunction [33]. Similarly, the myocardial blush score is another quantitative angiographic method of indirectly assessing coronary microcirculatory dysfunction. The predominant use of both CFTC and the myocardial blush score has been to assess the effectiveness of revascularisation in AMI patients [34,35]. However, a lack of reproducibility among many other limitations, has led to limited clinical use of these semiquantitative applications.

## 5. Coronary Microvascular Dysfunction: Can We Be More Precise?

From the results of the two above-mentioned key studies (ORBITA and ISCHEMIA), we can infer the existence of microvascular coronary dysfunction in many or all patients with SAP. Recent studies have focused intensely on the nature of these putative anomalies and have provided evidence that they constitute both structural (anatomic) and dynamic (physiological) components [36,37,38]. Both components have now been shown to contribute substantially to the pathogenesis of SAP and, indeed, of other forms of cardiovascular disease. The majority of investigations have focused, in particular, on the dysfunction of the coronary vascular endothelium, largely but not entirely, with regard to the endothelial modulation of vascular smooth muscle reactivity. However, recent investigations have extended beyond changes in vascular reactivity in isolation. There has been an increasing recognition that *structural remodelling of the endothelial glycocalyx*, an acellular, carbohydrate-rich layer separating the endothelial cells from the circulating blood, represents another key aspect of the pathogenesis of coronary microvessel dysfunction [39].

(a)
*Structural anomalies: Focus on the endothelial glycocalyx*


“Shedding” of the inner endothelial lining (glycocalyx) has been suggested as one of the causes of acute myocardial ischemia (AMI) and represents the major component of structural dysfunction^11^. “Shedding” of the glycocalyx has been studied mainly in the context of cardiac emergencies, such as takotsubo syndrome and AMI rather than SAP. Shedding of the glycocalyx is caused by “sheddases”, which are released via the destabilisation and degranulation of mast cells [40] The “sheddases” include multiple enzymes, such as multiple matrix metalloproteases, released especially from mast cells, platelets and leukocytes, and the ADAM family [41]. 

This enzymatic erosion of the glycocalyx potentially leads to increased vascular permeability, impaired shear-stress-dependent nitric oxide (NO) production, the induction of vascular inflammation and the impairment of vascular and vascular endothelial energetics, impaired vascular rheology, and increased propensity towards thrombosis [42,43], as depicted in Figure 3. In turn, disturbed vascular rheology increases microvascular resistance and reduces the ability to initiate endothelium-dependent coronary vasodilator responses [44]. Nonlaminar flow patterns also stimulate thioredoxin-interacting protein (TXNIP) expression and NLRP3-inflammasome activation [45]. This cascade promotes mitochondrial dysfunction and the resultant oxidative stress and impaired glucose utilisation, leading to endothelial cell apoptosis [46].

Most importantly, glycocalyx erosion increases the propensity toward platelet aggregation as a basis for eventual micro- or macrothrombosis independent of the presence of atheromatous plaques. Thus, recent studies of the thrombotic substrate in AMI have suggested that approximately 20–25% of AMI cases reflect superficial regional coronary endothelial erosion rather than ruptured atherosclerotic plaque [48]. 

There is increasing evidence to support the concept that glycocalyx shedding is a modulator both of chronic and acute myocardial ischaemia. The impact of glycocalyx shedding on chronic coronary microvascular physiology has thus far been studied only scantily at a clinical level. However, a recently published study [49] showed that among patients undergoing invasive investigation for suspected coronary artery disease, there is a direct correlation between the extent of the impairment of coronary microvascular dilator response to ATP and plasma concentrations of syndecan-1 (SD-1), a major component of the glycocalyx. Furthermore, both AMI [50] and crises of coronary artery spasm [51] are associated with the acute release of SD-1 into plasma (see Figure 4).

Moreover, if we apply this concept to the pathogenesis of SAP, it is very possible that glycocalyx shedding may also limit the coronary flow reserve, predisposing to myocardial ischemia. However, this has not been investigated thus far. Moreover, there is also no real body of work related to changes in the rates of glycocalyx shedding from the coronary vascular bed in chronic disorders, given that the only convenient methods for such studies would rely on documenting transvascular gradients of the glycocalyx components. Overall, an appropriate investigation of the association of glycocalyx damage with chronic myocardial ischemia would represent great technical difficulty because of the need to measure the erosion of glycocalyx specifically from the small coronary arteries. 

There is a close relationship between *platelet activation* and erosion of the glycocalyx. Fundamentally, platelets from patients with acute coronary syndromes release matrix metalloproteinase-2 (MMP-2), a major source of glycocalyceal damage [52,53], while, in turn, MMP-2 induces further platelet activation. Experimentally, the relationship between the induction of glycocalyx injury and propensity towards the coronary vascular adhesion of platelets was well demonstrated by Chappell et al. [54], who studied Langendorff-perfused guinea pig hearts subjected to ischemia/reperfusion. This process markedly increased the release of glycocalyx components into the coronary venous effluent, with a simultaneous 2.5-fold increase in platelet adhesion to the coronary vasculature. Intriguingly, there is also emerging evidence that damage to the glycocalyx induces monocyte activation [55]; thus, the combination of the impairment of the endothelial barrier function and the activation of monocytes contributes to the initiation of inflammatory activation in the vasculature and myocardium. Importantly, while there is evidence (as above) implicating glycocalyx damage as a stimulus for platelet activation, it is currently far from certain whether glycocalyx damage modulates vascular smooth muscle sensitivity to homeostatic autacoids, such as NO. This information is now of critical importance with regard to the genesis of a “pro-aggregatory cascade” in association with glycocalyceal damage.

Finally, there are increasing concerns that damage to the endothelium and, in particular, to the endothelial glycocalyx may result in irreversible impairment of the endothelial function even after repair mechanisms have resulted in the restoration of the anatomical integrity of the glycocalyx. For example, regenerated endothelium displays impaired capacity for nitric oxide (NO) release [56];

(b)
*Physiological anomalies: Focus on the endothelial dysfunction*


On the other hand, **physiological microvascular dysfunction** refers to the imbalance between the vasoconstrictor and vasodilator modulation of coronary artery tone, with a net diminution of the effects of vasodilator/antiaggregatory/anti-inflammatory autacoids. Particularly relevant to impairment is the diminution of the effects of NO, prostacyclin (PGI_2_), and hydrogen sulphide (H_2_S), as important endogenous coronary vasodilators, which potentiate each other’s effects, whether vasodilator, antiaggregatory, or anti-inflammatory. 

The majority of studies on the physiological dysfunction of the microvasculature focus on anomalies of NO signalling. Indeed, there are good reasons for this, both historical and pathophysiological. As Vanhoutte et al. observed in a recent review of vascular endothelial dysfunction [56], in the more than 40 years since Furchgott and Zawadzki [57] reported that stripping of the vascular endothelium resulted in impairment of vasodilator function, there has been a continuous process of the delineation of various anomalies leading to combinations of reduced rates of NO formation, as well as the impairment of the NO signalling cascade, which focuses on its “receptor”, soluble guanylate cyclase. The latter deficiency of end-organ responsiveness to NO is designated *NO resistance*. The major mechanism of NO generation is the conversion of arginine to NO and citrulline by three distinct tissue NO synthases, of which NOS3, or endothelial NOS, is most relevant to the vasculature. NOS activity may be modulated by the availability of cofactors, such as tetrahydrobiopterin (BH4), and by variability in the tissue concentration of endogenous NOS inhibitors, such as asymmetric dimethylarginine (ADMA). Once NO is released, this can be “scavenged” by the superoxide anion (O_2_^−^) under conditions of oxidative stress, leading to the generation of peroxynitrite, an indirect modulator of impaired cellular energetics [58]. Finally, soluble guanylate cyclase is readily oxidised, with or without an associated loss of its haem moiety, resulting in the reduction in formation of cyclic guanosine monophosphate (cGMP), which mediates most biological effects of NO [59]. There is even some evidence that soluble guanylate cyclase, when subjected to oxidative stress, may lose its selectivity for the production of cGMP, instead generating the vasoconstrictor inosine phosphate [60].

The other well-characterised microvascular coronary vasodilator of endothelial origin is *PGI_2_*, which is generated as a major product of the actions of cyclo-oxygenases on arachidonic acid. The effects of PGI_2_ are mediated via binding to specific (IP) receptors, which are coupled via G-proteins to the activation of adenylate cyclase [61]. Although PGI_2_ has attracted relatively little interest until recently with regard to the pathogenesis of microvascular dysfunction, there is increasing evidence of its physiological and pathophysiological significance. Interestingly, PGI_2_ release contributes substantially to the normal vasodilator effects of acetylcholine (ACh) [62]. Figure 5 compares the biochemical pathways responsible for the inhibition of platelet aggregation by NO and PGI_2_, including their eventual confluence and, therefore, synergy.

It is likely that there are several other vasodilator autacoids of endothelial origin, all of which exhibit synergistic effects with those of NO and PGI_2._ These include H_2_S, angiotensin (1–7), and carbon monoxide (CO), of which the first two are particularly relevant in the context of microvascular disease.

*H_2_S* was first identified as a vasodilator coronary autacoid approximately 20 years ago [63], and ongoing studies have served to emphasise the physiological importance of its homeostatic effects at many levels, including the regulation of vascular tone and thrombus formation. Many, but not all, effects of H_2_S are synergistic with those of NO [64,65,66]. Therefore, one potential cause of impaired responsiveness to NO is the reduced availability of H_2_S. Importantly, H_2_S also appears to modulate adenylate cyclase signalling [67] and, thus, may modify PGI_2_ signalling. Furthermore, H_2_S stabilises mast cells, which represent a major source of matrix metalloproteinase release and, thus, of platelet and glycocalyx activation [68] and inhibits activation of the NLRP3 inflammasome, thus protecting the endothelium from oxidative stress [69]. Overall, it can now be appreciated that the partial synergy between H_2_S, NO, and PGI_2_ represents the basis for the maintenance of the homeostasis of vasodilator mechanisms within coronary microcirculation.

At this stage, less is known about the physiological and pathophysiological roles of *angiotensin* (1–7)*,* the main product of the dipeptidase action of the angiotensin-converting enzyme type 2 (ACE2). Previous studies [70] have shown that angiotensin (1–7) potentiates the effects of NO. However, the physiology of ACE2 and of angiotensin (1–7) has been thrown into sharp relief by studies related to COVID-19-induced vasculopathy, which will be discussed below. Furthermore, emerging studies suggest that angiotensin (1–7) plays a major role in protecting the microvasculature from the impairment of NO-induced vasodilatation associated with the presence of coronary artery disease [71]. Intriguingly, this effect is mediated via increases in telomerase activity, suggesting an “ant-ageing” effect of angiotensin (1–7) in the microvasculature [72];

(c)
*Importance of vascular interactions with mast cell products and with circulating platelets*


*Mast cell activation*, as described above, represents a fundamental source of inflammatory change throughout the body [73], and the various vasodilator autacoids also maintain homeostasis via mast cell stabilisation [74]. Of the various proinflammatory agents released into the circulation after mast cell degranulation, histamine is notorious for inducing the impairment of the glycocalyx function and, thus, increasing vascular permeability [75]. Thus, mast cell activation should be regarded as an “upstream” source of both structural and physiological endothelial dysfunction.

It is important to recognise that all of the pathophysiological changes described above in the coronary microvessels have *parallels within circulating platelets*. Thus far, little has been written about disorders of the platelet glycocalyx function, but it is certainly clear that the limitation of platelet activation and induction of aggregation represents the result of synergistic interactions between autacoids, such as NO, PGI_2_, H_2_S, and angiotensin (1–7) [61]. Conversely, there is strong evidence that the impairment of antiaggregatory responses to NO (“nitric oxide resistance”) in platelets represents independent markers of poor prognosis [76]. It increasingly appears that platelet and vascular smooth muscle reactivity changes occur in tandem and represent distinct contributors to the risk of thrombotic events. In practice, the evaluation of platelet physiology, therefore, represents an indirect but convenient means of evaluating potential anomalies in vascular function in human studies. Figure 6 is a schematic representation of the factors modulating the mast cell–endothelial glycocalyx interaction.

## 6. Some Examples of Pathophysiological Changes within the Coronary Microcirculation

Having delineated the current understanding of the process involved in the pathogenesis of microvascular dysfunction, it is important to place these in a clinical context. Over the last 5 years, there has been an explosion of investigations into the importance of both the structural and physiological aspects of microvascular disease in a wide range of acute and chronic cardiovascular disease states, including but not limited to myocardial ischaemia. 

(a)Myocardial ischaemia

This review primarily addresses microvascular anomalies in the setting of various myocardial ischaemic syndromes. However, it is now clear that other conditions also interact with microvascular dysfunction;

(i) Stable angina pectoris (SAP)

Patients with SAP have previously been thought to represent a clinical syndrome derived entirely from the constraints of associated “haemodynamically significant” large-vessel coronary stenoses. However, this concept has been changed, not only by the results of the ORBITA study [16] but by the frequent occurrence, especially in women, of exertional angina occurring in the absence of large coronary artery stenoses. A finding of “normal coronary arteries” is typically made in something of the order of 25% of patients coming to diagnostic coronary angiography despite typical exertional anginal symptoms, and the condition has been labelled “INOCA” (ischaemia with nonobstructive coronary arteries) [77]. The factors associated with the occurrence of INOCA in the ISCHEMIA trial are summarised in Table 2. A number of studies have documented the presence of impaired coronary vasodilatation, typically with the intracoronary infusion of adenosine and/or acetylcholine in such patients (see [78] for review). In many such cases, a diagnostic coronary arteriography revealed minor plaque formation in large coronary vessels, raising the possibility of a mechanistic association with atherogenesis. However, the main demographic association with INOCA has been the female sex. Aziz et al. [79] documented a female-to-male ratio of 4.2 after multivariate analysis. Similarly, in the ISCHEMIA trial, 13% of patients who were enrolled on the basis of exertional angina and investigative evidence of inducible myocardial ischaemia were not randomised because of the absence of severe large-vessel stenosis [80]. These individuals tended to be younger and more often were women compared with patients with stenotic large-vessel disease. Furthermore, among patients with SAP, the majority exhibit the impaired antiaggregatory effects of nitric oxide [81] and antiaggregatory prostanoids [61,82];

(ii) Prinzmetal’s angina/coronary slow flow phenomenon (CSFP)

It has been apparent for many years [83] that many patients experience episodes of angina pectoris, which are largely independent of exertion and rather occur at night in a cyclical and largely unpredictable manner. Such episodes of angina are often prolonged and may be precipitated by emotional stressors. The cause of this “variant” form of angina has been shown to be an episodic coronary artery spasm (CAS), essentially independent of the presence of coronary atheromatous plaques. Interestingly, episodes of CAS are notorious for responding poorly to sublingual nitroglycerine, suggesting impaired NO responsiveness. A spasm in large coronary arteries can be either focal or diffuse and is usually called Prinzmetal’s angina (PA). A second form of CAS affects mainly or entirely the small coronary arteries, resulting in the development of the slow coronary flow phenomenon (CSFP), whereby coronary flow rates are markedly retarded in some or all epicardial coronary arteries [84]. We have previously shown that CSFP can be reversed and symptoms in CSFP improved by treatment with mibefradil, a selective blocker of T-type microvascular calcium channels [85]; thus, CSFP arises from microvascular CAS. 

We recently also demonstrated [51] that patients with CAS exhibit severe resistance to the antiaggregatory effects of NO, which is exacerbated during symptomatic crises. These crises were also associated with the release of the glycocalyx component syndecan-1 (SD-1) into circulating blood, together with an increased generation of platelet microparticles and increased concentrations of the mast cell enzyme tryptase in plasma. N-acetylcysteine reversed all of these changes via the breakdown of cysteine and the release of H_2_S.

It is, therefore, likely that CAS largely reflects simultaneous platelet activation and glycocalyx “shedding”, triggered by the destabilisation of mast cells. The role of NO and H_2_S signalling impairment as a key aspect of homeostatic impairment in CAS is clearly a central one.

(b)Normal Ageing

Patient age has long been characterised as an independent risk factor for thrombotic events such as AMI and thrombo-embolic complications of atrial fibrillation, but the reasons for this association are only beginning to become apparent. More recently, a number of proteins have emerged as modulators of the cardiovascular ageing process. Interestingly, some of these appear to interact with the autocidal control of the microcirculation. In many respects, the vascular ageing process can be regarded as being particularly driven by changes within the vasculature, particularly arterial stiffening, which is partially a fibrotic and partially a vasomotor disorder.

As with other aspects of microvascular dysfunction, it is emerging that there are often parallel changes in both vascular and platelet reactivity in association with normal ageing, and a number of determinants of premature vascular ageing have been identified.

One of the major alterations that occurs with normal ageing is the shortening of chromosomal telomeres, limiting the capacity for normal chromosomal replication (reviewed by Razgonova et al. [86])*. Telomerase* is an enzyme complex, which counters telomere shortening by adding nucleic acids to the ends of chromosomes. This alone constitutes an antiageing effect. However, telomerase also appears to interact with mitochondrial function and to protect the functionality of nitric oxide-mediated arterial dilatation by preventing a shift from NO-mediated to hydrogen peroxide-mediated dilatation [87]. Furthermore, at least part of the potentiation of NO-mediated vasodilatation by angiotensin (1–7) is mediated by increases in telomerase activity [71].

Another important “anti-ageing” protein is *Klotho*, named after the thread-spinning fate sister of Greek mythology. Klotho is generated largely within the renal parenchyma (as a transmembrane protein) but is partially released into plasma, where it remains active. Klotho appears to represent a fundamental antiageing protein (reviewed by Donato et al. [88]), especially with regard to the limitation of age-related vascular stiffening and the loss of endothelial function. Decreases in plasma concentrations of free Klotho can be taken as a surrogate of accelerated vascular ageing, related to its effect in countering oxidative stress within the vasculature [89]. Indeed, levels normally decrease with age. Decreased Klotho concentrations correlate not only with impaired renal function but also with apparent arterial stiffness and with systolic blood pressure. There is increasingly strong evidence that Klotho protects the vascular endothelial function, especially in patients with renal dysfunction [90]. There is some evidence that H_2_S may act (via the suppression of the ACE-induced generation of angiotensin I) to increase plasma concentrations of Klotho in ageing individuals [91].

Against this still-uncertain theoretical basis for ageing-related vascular stiffening and vascular dysfunction, there is strong evidence that NO’s effects on its target tissues diminish with normal ageing [92] and that this deterioration is sharper in women than men. Furthermore, the decreased antiaggregatory effect of NO in platelets is correlated with an increased expression of the proinflammatory protein thioredoxin-interacting protein (TXNIP), which may theoretically contribute to increased oxidative stress within platelets (and blood vessels) [93];

(c)Diabetes Mellitus (DM)

Patients with diabetes have increased short-and long-term risks of both macrovascular and microvascular complications, especially if there is substantial long-term hyperglycaemia. It has been documented for many years that diabetes, even in the absence of marked hyperglycaemia, is associated with biochemical and physiological markers of inflammation, oxidative stress, and vascular endothelial dysfunction, both of the macrovascular and microvascular types [94,95]. Furthermore, we have previously shown that the combination of myocardial ischaemia and uncontrolled diabetes, associated with severe hyperglycaemia, leads to superoxide release into plasma, impairment of NO formation (probably via the prolongation of ADMA half-life, presumably via inactivation of its enzymatic metabolic processes), and of NO signalling at the platelet level [96].

The main subject of recent progress in this area is the involvement of glycocalyx degradation in the pathophysiology of diabetic microangiopathy, even though this has not thus far been investigated specifically at the coronary level. Heparanase and hyaluronidase, both of which may act as glycocalyx “sheddases”, have been implicated in vasculopathy in both animal models and humans. Clinically, patients with both type I and type II diabetes have elevated plasma concentrations of glycocalyx components [97]. These data argue strongly for a new focus on the maintenance of glycocalyx integrity in the management of diabetic patients;

(d)Heart failure with preserved ejection fraction (HFpEF)

HFpEF is a syndrome of impaired exercise tolerance induced by the failure of the heart to adequately adjust its haemodynamic parameters to exercise; for example, the cardiac output does not increase in parallel with the myocardial oxygen demand. HFpEF is defined largely by exclusion: the left ventricular ejection remains normal, at least at rest, and there is no substantial large-vessel ischaemia limiting the exercise capacity. As ageing, hypertension, and obesity all predispose towards the development of HFpEF, the prevalence of this condition has increased sharply in recent years in most societies, and it now accounts for more than half of the cases both of heart failure and of heart failure hospitalisations [98]. Although the pathophysiology of HFpEF is likely to be somewhat heterogeneous, it has attracted considerable interest over the past 10 years because of the recognition that, until very recently, there was no form of heart failure treatment which had any beneficial effect on either the symptomatic status or prognosis of HFpEF patients.

In 2013, Paulus and Tschope [99] proposed that HFpEF had a number of distinct pathogenetic components, including a systemic proinflammatory state, coronary microvascular endothelial inflammation, and impaired NO signalling with consequent impairment of myocardial relaxation. As studies of human myocardial biopsies have emerged, these ideas have acquired direct supporting evidence. However, and somewhat paradoxically, it has also emerged that HFpEF may be driven, in part, by nitrosative stress, which is the formation of peroxynitrite representing a combination of increased inducible NO synthase (iNOS) expression and the generation of a superoxide anion due to oxidative stress [100]. This finding is particularly important because peroxynitrite indirectly activates the “energy sink” enzyme poly(ADP/ribose) polymerase (PARP), which causes myocardial energetic impairment and poor LV relaxation, such as occurs in HFpEF;

(e)COVID-19-induced vasculitis.

During the earliest stages of the worldwide spread of the SARS-CoV-2 virus, which causes the COVID-19 infection, it was not only apparent that the virus could interfere with normal functioning of multiple organ systems but that these included the heart, lungs, kidneys, and brain. It also rapidly emerged that the vasculature was a target for COVID-19, and a number of investigators hypothesised that COVID-19 was fundamentally a disease of the vascular endothelium (for review of this hypothesis, see [101,102]). The argument for a fundamentally vascular pathophysiology of COVID-19 was enhanced when it was found that the primary method for cell penetration of the virus was the binding of its spike protein to ACE2 [103], the enzyme responsible for the generation of angiotensin (1–7) and one that is particularly abundant within the coronary vasculature and heart [102]. It has also been reported that following COVID-19 virus entry into cells in mice models, there is the downregulation of ACE2 expression [104] and that this might potentially result in impaired myocardial contractility [105].

The finding of decreases in plasma ACE2 expression and activity post-COVID-19 appears to be substantially important. For example, Mongelli et al. [106] reported a substantial decrease in ACE2 expression in peripheral blood in all patients studied after COVID-19 infection. This was combined with a substantial increase in telomere shortening and in the calculated biological age. 

On the other hand, in populations with no history of COVID-19 infection, *increased* circulating ACE2 concentrations, which largely reflect release from damaged endothelial cells [107], have now been demonstrated to be an independent marker of cardiovascular and mortality risk [108]

A critically important knowledge deficiency relates to the relationship between the recovery of the vascular endothelium (and especially the endothelial glycocalyx) and the normalisation of both ACE2 expression and its release into the circulating blood. Such studies offer the potential to better characterise the “long COVID” syndrome in a quantitative rather than categorical and symptom-based and categorical rather than quantitative manner.

## 7. Therapeutic Implications

Given that it is now evident that the coronary microcirculation represents a major modulator of coronary perfusion and that inflammation of the coronary microvessels (especially their endothelial layers and glycocalyces) modulate not only symptomatic myocardial ischaemia but also the cardiovascular ageing process, diabetic heart disease, HFpEF, and COVID-19-related coronary vasculitis, it is appropriate to consider the development of new prophylactic and therapeutic options to reduce the population health burden from these and, in all probability, other disease states.

From a theoretical point of view, new agents should be developed with a view especially towards stabilising the endothelial glycocalyx and the effect of endothelial vasodilator/antiaggregatory/anti-inflammatory autacoids.

From a theoretical point of view, the development of these new agents should be seen to improve rheology in small coronary vessels, reduce vascular permeabilisation under oxidative stress, improve energetic status in microvessels, platelets, and myocardium, and decrease myocardial infiltration with monocyte/macrophages and neutrophils. There needs to be a reduced propensity towards platelet activation and aggregation without the induction of a bleeding diathesis. 

In terms of the benefits from a clinical physiology point of view, the requisite processes need to include a re-evaluation of Gregg’s phenomenon as a specifically microvascular therapeutic target [59]. Specifically, the concept that there may be an interaction between the coronary perfusion rate and myocardial contractile performance is of potential therapeutic interest in all of the conditions which have been discussed above.

It is also likely that a research approach directed at achieving combined optimisation of the effect of homeostatic autacoids may be therapeutically useful, not only for the prevention of paradoxical coronary vasoconstriction but also to stabilise mast cell and platelet function and, thus, prevent ischaemic crises. In this regard, the potential utility of agents which increase H_2_S release is a particularly promising area, not only because of studies with N-acetylcysteine, which has limited oral bioavailability, but also with other sulphydryl compounds [109]. Furthermore, the potential role of donors of H_2_S via cysteine conjugates of sources of polysulphides for the prophylaxis and treatment of coronary microvascular disorders has recently been enhanced by a series of discoveries related to the important accessory properties of H_2_S (beyond intrinsic vasodilatation, the inhibition of aggregation, and the potentiation of the effects of NO). Specifically, these “new” effects include the antagonism of some T-type calcium channels [110] and the activation of adenylate cyclase-mediated signalling but simultaneously exerting a dual role in this system, in suppressing preactivated (for example, by prostacyclin) adenylate cyclase signalling [69]. On these and other bases, hydrogen sulphide donors have been heralded as key emerging preventive and therapeutic agents for ageing-related disease states.

It is appropriate to end this section on potential therapeutics with a brief mention of the emerging beneficial effects, in microvascular disorders such as HFpEF, of agents such as the sodium-glucose. Originally introduced as novel treatments for type 2 diabetes, they were found during routine cardiovascular safety testing to exert beneficial effects on the composites of heart failure admissions and deaths, and they now represent the first known beneficial agents for the treatment of HFpEF [111]. Juni et al. [112] reported that the SGLT2 inhibitor empagliflozin enhances cardiomyocyte systolic and diastolic function by preventing the impairment of the endothelial cell–cardiomyocyte interaction which protects cardiac contractility, an effect which suggests restoration of a normal Gregg’s phenomenon. Furthermore, empagliflozin improves cardiac energetics and preserves left ventricular systolic function in a porcine model of heart failure, associated with a shift away from glucose in myocardial substrate utilisation [113]. Although there is much to be learned about the mechanisms and therapeutic extent of these beneficial effects, further investigation of SGLT2 inhibitor effects is clearly warranted. Table 3 summarises our current understanding of potential avenues for the therapeutic protection of the endothelial glycocalyx and, thus, coronary microcirculation.

## Figures and Tables

**Figure 1 ijms-24-11287-f001:**
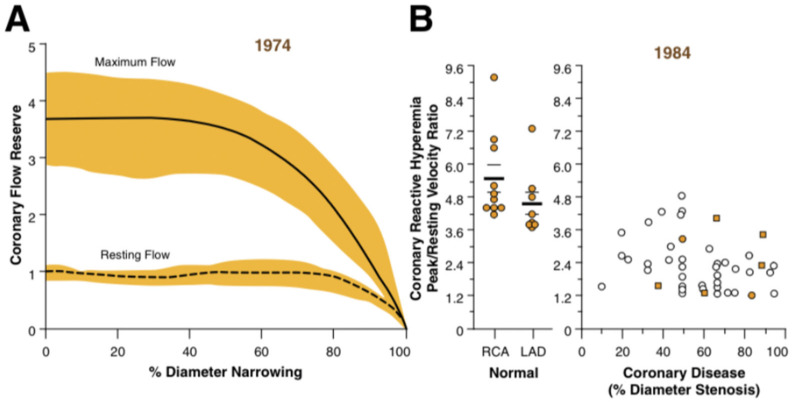
Evaluations of structural determinants of coronary flow reserve in (**A**) a canine model and (**B**) patients undergoing coronary bypass surgery, with data related to single large coronary artery stenoses. Note disparities between stenosis–flow reserve relationship in (**A**) versus (**B**). Reproduced, with permission from Gould KL, *JACC Cardiovascular Imaging*, 2009 [2].

**Figure 2 ijms-24-11287-f002:**
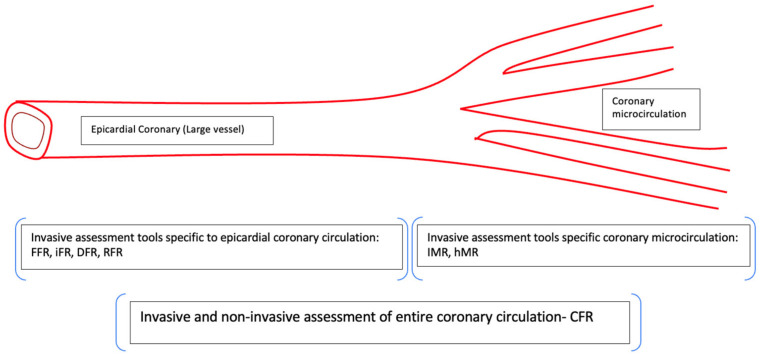
Schematic representation of large and small coronary vasculature, with corresponding investigations to examine the functional status of each region. Invasive physiological assessments tools can be specific for macrocirculation (FFR and nonhyperemic indices—iFR, RFR, DFR) and microcirculation (IMR and hMR) or can assess coronary circulation as a whole (CFR). CFR = coronary flow reserve, DFR: diastolic hyperaemic free ratio, FFR = fractional flow reserve, hMR = hyperaemic microvascular resistance, iFR: instantaneous wave-free ratio, IMR: index of microvascular resistance, RFR: resting full-cycle ration.

**Figure 3 ijms-24-11287-f003:**
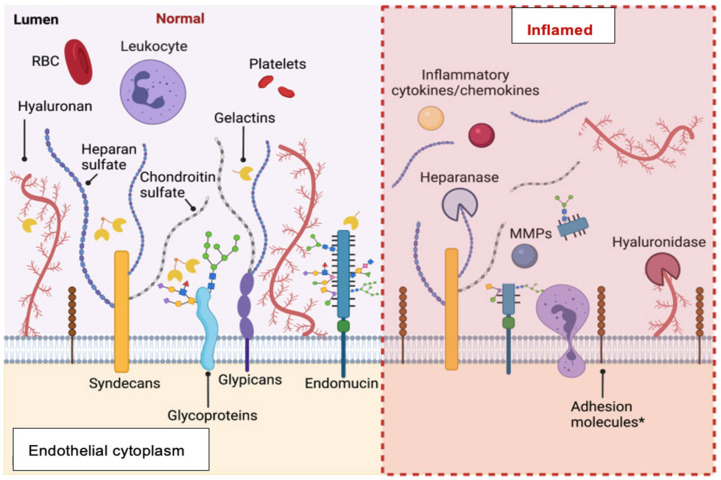
Determinants of turnover of components of the endothelial glycocalyx under resting and inflammatory conditions. Note continuous interactions with circulating blood cells even under resting conditions and chemical bases for glycocalyx “shedding” under inflammatory conditions. Modified, with permission, from Hu et al., *Frontiers in Cell and Developmental Biology*, 2021 [47].

**Figure 4 ijms-24-11287-f004:**
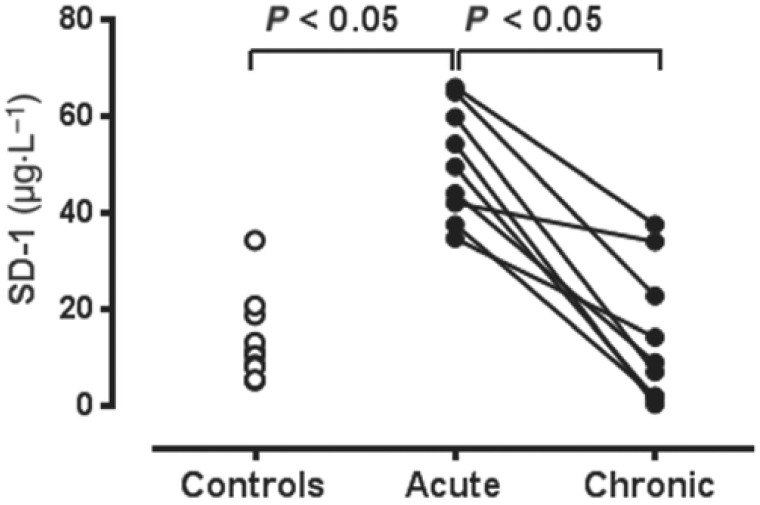
Plasma concentrations of the glycocalyx component syndecan-1 in normal (“control” subjects compared with concentrations in patients with known coronary artery spasm during acute and chronic phases). Reproduced with permission from Imam-H et al., *Br. J. Pharmacol.*, 2021 [51].

**Figure 5 ijms-24-11287-f005:**
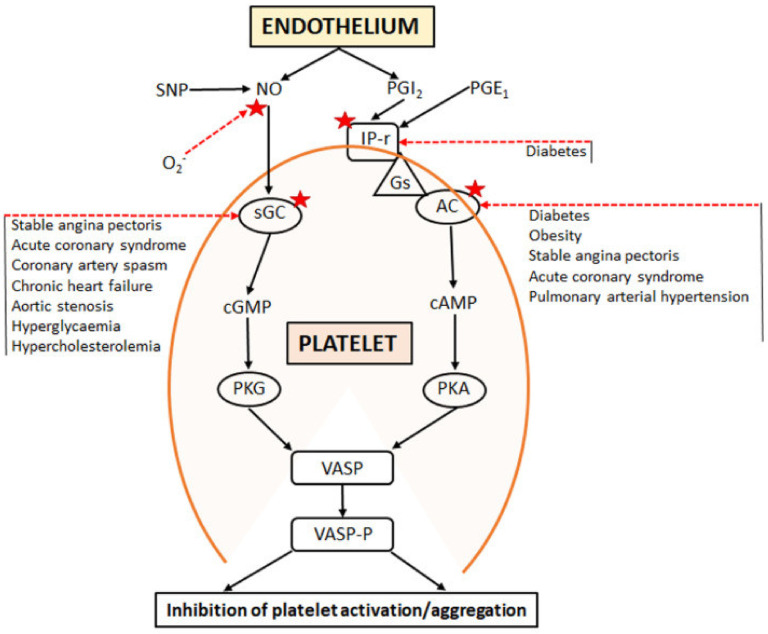
Biochemical “cascades” controlling integrity of nitric oxide (NO) and prostacyclin (PGI_2_) signalling individually and in combination in respect to homeostasis within endothelium and platelets. Note associations of signalling defects (★) with a variety of cardiovascular disease states. Reproduced with permission, from Chirkov-YY et al., *Int. J. Mol. Sci.*, 2022 [61].

**Figure 6 ijms-24-11287-f006:**
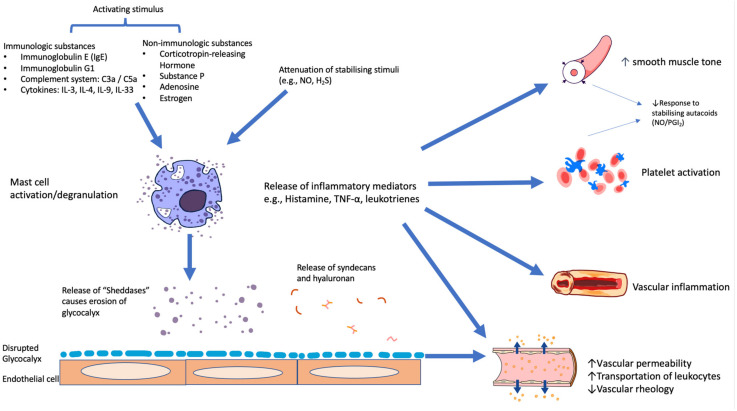
**Schematic representation of** modulators and mediators of effects of mast cell activation: the potential impact of mast cell degranulation on glycocalyx damage and resultant microvascular inflammation, increased microvascular permeability, impaired microvascular autocidal function, and platelet activation/aggregation.

**Table 1 ijms-24-11287-t001:** Cardiac events (total numbers) in patients randomised to invasive or conservative management in the ISCHEMIA trial, as reported by Lopez-Sendon et al. (2022), after mean of a 3.2-year follow-up [18].

Event (*n* = 2591)	Conservative Group (*n* = 2591)	Invasive Group (*n* = 2588)
Primary endpoint	451	420
Death	144	145
Myocardial infarction	268	244
Heart failure	33	62
Unstable angina	35	17

**Table 2 ijms-24-11287-t002:** Clinical factors associated with myocardial ischaemia in the absence of coronary artery stenoses (INOCA); data from Reynolds et al., *JACC Imaging*, 2022 [80].

Clinical Parameter	*p*-Value	Odd Ratio
Increasing age	<0.001	0.77
Female	<0.01	4.19
Diabetes	<0.01	0.56
Cigarette smoking	NS	0.68

Data shown generated using multivariable analyses. Note dissociation between likelihood of INOCA and “conventional” coronary risk factors.

**Table 3 ijms-24-11287-t003:** Potential therapeutic options to limit erosion of the endothelial glycocalyx.

Category	Examples	Potential Clinical Utility
Mast cell stabilisation: (1) Autacoids (2) Exogenous agents	Nitric oxide (NO), prostacyclin (PGI_2_) Hydrogen sulphide(H_2_S) Corticosteroids Trimetazidine	Both NO and PGI_2_ may inhibit mast cell activation and degranulation. However, no clinical studies to date have conclusively shown the benefit of exogenous administration of such agents. The H_2_S donor N-acetylcysteine, in association with infused nitroglycerine, has been shown to reduce myocardial infarct size [50] and to stabilise crises of coronary artery spasm [51], with associated reduction in plasma tryptase concentrations. There are isolated case reports documenting reversal of coronary spasm crises with high-dose corticosteroid “pulse” therapy. Trimetazidine, which is known to act as a partial fatty acid oxidation inhibitor, has an established antianginal effect. This may be mediated, in part, by increased formation of H_2_S [114].
Inhibition of “sheddase” (largely matrix metalloproteinase) effect in degrading glycocalyx	(Low-dose) Doxycycline	Low-dose doxycycline, while exerting no antibiotic effect, acts as a nonspecific inhibitor of matrix metalloproteinases. However, no trials to date have convincingly established its utility to prevent coronary events. A number of other candidate MMP inhibitors are under development [115].
Increased glycocalyx regeneration	Albumin	Albumin is the major plasma protein and is responsible for the majority of plasma oncotic pressure. In vitro studies have shown that glycocalyx regeneration, especially in diabetic models, is accelerated by albumin. However, plasma albumin repletion has not been utilised therapeutically to date.
Reversal of endothelial inflammation	SGLT2 inhibitors	Two studies [112,116] have shown inhibition of microvascular inflammation by SGLT2 inhibitors. A similar effect was seen with N-acetylcysteine [116], but full mechanism(s) is not yet delineated.

## Data Availability

All data for this review article were extracted from previously published articles.

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
