# Peer review of "Coronary “Microvascular Dysfunction”: Evolving Understanding of Pathophysiology, Clinical Implications, and Potential Therapeutics"

_ijms, 2023, doi:10.3390/ijms241411287_

Round 1
Reviewer 1 Report
The authors have undertaken a comprehensive review of microvascular dysfunction. they start off with the symptom of chest pain and demonstrate how the current literature now points strongly towards the role of microvascular dysfunction. They provide an analysis of the role of FFR and the FAME studies and extend to the modern ORBITA and ISCHAEMIA studies. Their contention is that large vessel stenosis and PCI have a limited role in stable angina and this well argued.
Next the authors move on to current tools for the academic and clinician to assess coronary microvascular function. They bring in a depth of understanding of physiology and link this with invasive and non-invasive methods.
They then develop a crucial section on the evolving evidence for the role of the glycocalyx in endothelial function platelet aggregation. This will be less familiar to many readers and represents a strength of this review. There follows some conjecture where the authors extrapolate from acute to chronic physiology where the data is lacking. There then follows a section on physiological microvascular dysfunction with a focus on signally pathways determining vasomotor tone in the coronary microvessels. The authors have particular expertise in platelet function and place a significant emphasis on vascular interactions with mast cell products and circulating platelets. Although this is written as a review, the authors advance their own hypothesis about platelet glycocalyx function with very limited direct evidence in support. Their proposal that platelet physiology represents an indirect but convenient means of evaluating potential anomalies in vascular function is provocative and perhaps oversteps the role of a review article.
They then go on to look at specific examples of coronary microvascular dyfunction including myocardial ischaemia (stable angina pectoris and coronary slow flow phenomenon), normal agining, diabetes mellitus, heart failure with preserved ejection fraction and COVID-induced vasculitis. This section is of interest but doesnt seem to flow from the pathophysiology section beyond the role of glycocalyx degradation in diabetic micro-angiopathy.
there is a final section on therapeutic implications. This again is less of a review and more using limited evidence to develop a hypothesis of future avenues for research.
I think the authors have tried a very difficult (but important!) task of linking novel basic science developments with an analysis of contemporary data for the treatment of cardiac chest pain. I think less emphasis on the large clinical trials and a clearer linkage between glycocalyx dysfunction and the platelet-endothelium interaction in the described disease states would improve the readability and message of the review.
Specific comments:
Page 1 abstact: 2nd last sentence has poor grammar (repetitiion of towards)
Page 1 introduction 1st sentence: poor grammar ("reflects imbalance OR impaired..."
Page 2 2nd paragraph. Inconsisdtency between vessel sizes is confusing. > 0.5mm and later close to 300microm...
Suggest condense section 2 page 2 and section 3 page 3
Page 5 para 2: inconsisten use of abbreviation IMR-Angio and Angio-IMR
Page 7 1st para: could present the assessment of glycocalyx damage as a challenge to be solved rather than a great technical difficulty
Page 10: 2nd sentence form end does not make sense.? remove
Page 11: section a final sentence. Could speculate on what is the mechanism/trigger for the loss of NO and H2S signalling
Page 11 section B aging: I'm not convinced that this section adds substantially to the review.
Page 13 section d HFpEF: Not really much of a review in this section. For consistency it would be useful to review the evidence (or lack) for glycocalyx dysfunction in HFpEF.
Page 14: section e1st sentence "...it was NOT only apparent..."
Page 14 section e: the authoirs should attempt to reconcile the opposite findings regarding ACE2 concentrations and prognosis in COVID and non-COVID populations.
Page 15 para 3: 1st sentence too long and needs rephrasing
Page 16: is there any evidence of effect of SGLT2 inhibition on glycocalyx? eg DOI: 10.1161/JAHA.119.015716
There are frequent spelling and punctuation errors which need correcting
Author Response
First, we wish sincerely to thank the reviewer for his appreciation of the "review" which we have undertaken. We must agree that he is correct in that our inclusion of very recent and evolving experimental data has the effect of making this less than a pure, traditional review. However, we submit that this is a very important area in evolution, and therefore we feel justified in including some emerging data. We will comment on all individual aspects.
(1) Issue of other physiological processes linked to acute/chronic glycocalyx "shedding": platelet activation/aggregation and loss of homeostatic autacoida function in vascular smooth muscle: This is of course a critically important area: we mean to imply that glycocalyx "shedding" sets off a sort of endovascular "tsunami", via activation of platelet aggregation with simultaneous attenuation of vascular responses to homeostatic autacoids such as nitric oxide. Of course, platelet pathophysiology is a longstanding interest of urs, but here we are on relatively firm ground. We have added an extra reference (number 51 in the modofoed text) which proves that glycocalyx damage leads to increased platelet adhesion to the coronary microvessels. As for loss of autacoidal homeostasis, that has been investigated only with acetylcholine thus far, and so it is not yet certain whether NO synthesis and/or signalling is the main problem. Nevertheless, the area seems well worth mentioning.
(2) The transition from pathophysiology to specific clinical examples is "rough". We agree that this transition was far from smooth, and have added a short paragraph so that the sequence can be seen to be logical.
(3) Minor/grammatical changes: These have been addressed in the modified manuscript.
Reviewer 2 Report
The abstract describes a recent shift in thinking about the causes of stable angina pectoris (SAP) and acute myocardial infarction (AMI), which were previously believed to be primarily caused by epicardial coronary artery stenosis due to plaque buildup. However, recent studies suggest that abnormalities in the coronary microvasculature are often present, even in the absence of large coronary artery stenosis. These structural abnormalities are believed to be induced by inflammation, which leads to erosion of the endothelial glycocalyx, disrupts laminar flow, and facilitates endothelial-platelet interactions. In addition, physiological dysfunction is closely linked to structural remodelling and is characterized by impaired responsiveness to coronary vasodilators, predisposing individuals to coronary vasoconstriction and micro-thrombus formation. These findings suggest the need for new directions in medical therapeutics for SAP and other cardiovascular disorders.
the review is timely, well structured, and emphasizes the importance of the vascular endothelium and endothelial dysfunction in coronary disease and other pathologies. However, from the point of view of format, I observe anomalies, for example, a change of letters and paragraph format from page 5. Table 2 does not have a table structure, it must be incomplete and on page 15 a table x is named, to which it is not known what it refers to. All this should be corrected before publication.
Author Response
We wish to thank the Reviewer for his appreciation of the manuscript and his recognition of the importance of the subject matter.
In the modified version of the manuscript, we have attended to the varius errors in the original text: we are grateful for their being pointed out, but embarrassed that they were present!
Reviewer 3 Report
Review of the manuscript no. ijms-2380201 entitled "Coronary microvascular dysfunction: evolving understanding of pathophysiology, clinical implications and potential therapeutics"
Coronary microvascular disease (CMD) belongs to the subset of disorders affecting the structure and function of the coronary microcirculation and is associated with increased risk of adverse events. In the reviewed manuscript Authors described the pathophysiology of CMD, techniques for diagnosis of this dysfunction, clinical manifestations of CMD and potential methods of CMD treatment. However, this article does not bring any new knowledge implications, and this topic has already been discussed in many earlier articles, for example in the article: Taqueti VR, Di Carli MF. Coronary Microvascular Disease Pathogenic Mechanisms and Therapeutic Options: JACC State-of-the-Art Review. J Am Coll Cardiol 2018;72:2625-2641.
Other important comments on the article:
1. Whole article was prepared very carelessly. Authors should standardize the font type and normalize the bolded fragments of the text.
2. Most of the cited references were published more than 10 years ago. The cited literature should be actualized.
3. There are errors in the order of citation of articles: after [5] is cited [7], lack of position [6]. After [19] are cited [22,23], lack of [20,21].
4. The figures, tables are not appropriate:
· The figures and tables are not the result of the Authors' own analyses, but are a copy of those that appeared earlier in other articles. Therefore they should be removed from the article. Only Figure 3 can be left in the article, but Authors should attach editorial approval concerning copy of this figure.
· There is lack of Table 2 in the article.
· Figure 2 is illegible and should be corrected.
· Tables and figures should be placed near their first reference in the text, not at the end of the paper.
· The abbreviations used in Figure 5 should be explained in the included legend.
· What does “Table x” (page 15) mean in the text of article?
· Citations of tables are insufficient. Authors cite Table 3 (page 10), than Table 2 (page 16). There is lack of citation of Table 1.
The article does not qualify for publication in the IJMS journal.
Author Response
We were disturbed to read the comments from Reviewer 3. Fundamentally, we believe that, apart from some of his/her comments on errors and omissions in the test (also noted by the other Reviewers), his comments are completely inappropriate. We will elaborate on this view.
(1)"This article does not bring any new knowledge implications... Taqueti and di Carla"
We are bemused by these comments, especially as the Reviewer has failed to discuss specifically any of the momentous developments in understanding of small coronary pathophysiology, and especially that of the endothelial glycocalyx, which have emerged since 2018. It is particularly galling that he refers to the Position Paper of Taqueti and di Carla, approopriate as it was at the time, as having any significant relationship to the material which we have discussed in our manuscript. (After all, Reviewer 1 has suggested that our work verges on being too avant garde.) In the absence of ANY comments about the scientific content of our review, it is hard to discern that the reviewer has read anything except the grammar.
The idea that in a REVIEW one should not include (properly cited) previous publications which have materially advanced recent knowledge is bizarre.
While, as stated here, the comments of Reviewer 3 are unique in our experience, we are grateful for his comments on inappropriate order of references, an unattached table etc. These mistakes are now corrected.